organic chemistry

fluorescent crystal, sodium-fluorescent crystal, blue fluorescence

**Author for correspondence:**
Dongzhi Yang
e-mail: dongzhiy@xzhmu.edu.cn

This article has been edited by the Royal Society of Chemistry, including the commissioning, peer review process and editorial aspects up to the point of acceptance.

# A novel sodium-fluorescent crystal

Yunsu Ma, Yongjie Liu, Yuan Wang, Fan Zhang and Dongzhi Yang

School of Pharmacy, Xuzhou Medical University, Xuzhou, Jiangsu 22004, People's Republic of China

YM, 0000-0003-2204-9094; DY, 0000-0001-6796-1613

In this work, a novel sodium-fluorescent crystal (Na-FS) was synthesized from 4-dimethylaminobenzoic acid and sodium hydroxide by one-pot hydrothermal method. The structure and conformation of Na-FS were confirmed by single-crystal X-ray diffraction and scanning electron microscope, and the optical properties were studied by fluorescence spectrometer. The results showed that: Na-FS was a triclinic crystal, space group was P-1, cell parameters $a$, $b$ and $c$ were 10.5113(3), 15.9198(5) and 15.9560(5) Å, respectively, and the number of independent atoms Z in a structure cell was two. Additionally, Na-FS has a blue fluorescence emission (around 360 nm under excited at the range of 230–300 nm) with great photostability and photobleaching resistance, and the quantum yield of Na-FS is 30.58%.

## 1. Introduction

As a new kind of functional material, metal-organic coordination polymer had been widely used in the fields of adsorption, light energy and catalysis, energy storage, medicine and so on [1,2]. Organic luminescent materials can be divided into organic small molecule compounds and organic compounds according to their molecular structure [3]: polymer, metal organic complex [4]. Among them, the chemical structure of organic small molecule compound was easy to adjust by introducing unsaturated groups such as double bond, benzene ring and various chromophores, the co-roll path of molecules can be changed, so as to change the wavelength of fluorescence emission of the material [5]. The fluorescence emission wavelength of organic small molecule compounds can be overlaid covering the whole visible light range, and the fluorescence quantum dots of organic small molecule compounds were highly efficient and easy to purify. However, the small molecule compounds were easy to crystallize, fluorescence quenching and stability were relatively poor [6]. Organic polymers had good thermal stability, but it was not easy to improve the purity. The properties of

metal organic complexes had advantages of organic and inorganic, which included the high fluorescence efficiency of organic compounds and good stability of inorganic compounds [7].

In coordination compounds, metal organic complexes are a novel inorganic–organic hybrid material, which are organic–inorganic hybrid materials with intramolecular pores formed by self-assembly of organic ligands and metal ions or clusters through coordination bonds [8]. In metal organic complexes, the arrangement of organic ligands and metal ions or clusters had obvious directivity, which can form different frame pore structures, thus showing different adsorption properties [9–11], optical properties [12–14], electromagnetic properties [15,16], etc. Due to the advantages of high sensitivity, short response time and low cost, fluorescence sensing has attracted extensive attention of researchers in recent years, including organic dyes, quantum dots, fluorescent proteins and other fluorescent materials have been designed and developed [17–20]. These fluorescent dyes had many advantages, such as high absorption coefficient, relatively strong fluorescence signal, great resistance to photobleaching, long fluorescence life and so on, which to some extent increase their sensitivity in the application and avail their application in biological aspects [21]. As known, fluorescent imaging had been recognized as a facile and powerful tool for the detection of biologically relevant species with the ability to visualize morphological details and monitor various physiological processes in living systems [22–24]. In the determination of analytes, compared with other detection methods, fluorescence detection has the characteristics of fast response and low detection limit [23].

With the progress of science, technology and the needs of human beings, researchers urgently need to develop new fluorescent materials. Fluorescent metal organic complexes are a rapidly developing crystalline material, which provides greater possibilities for the exploration of new fluorescent materials and has great application potential. Fluorescent metal organic complexes have luminescent characteristics due to the diversity of their structures. The first report on luminescent metal organic complexes appeared in 2002. After that, many articles reported the luminescent properties of metal–organic frameworks (MOFs), and some reviews on the luminescent properties of metal organic complexes were published on some aspects [25,26], such as: separation [27], catalysis [28], biomedicine [29], sensing [30] and luminescent devices [31], etc.

Metal organic complexes as a new kind of luminescence material, had the adjustable fluorescence performance due to the abundant luminescence centre function sites [32], The fluorescence properties can be regulated by changing the metal centre, ligand and guest molecules, which makes metal–organic frame materials have great potential in the field of fluorescence sensing materials [33–36]. New fluorescent materials provided more possibilities for sensing or imaging application in many fields. It is essential to develop novel materials to extend the family of fluorescent materials.

Benzoic acid and its derivatives, as ligands of organic carboxylic acids, had been widely used in the synthesis of coordination polymers [37]. Not only because of the rich coordination patterns of them, but also they could be modified by different groups [38–40]. Among them, phthalic acid, chlorophthalic acid [38], terephthalic acid [39] and homophenzoic acid [40] had been used to synthesize metal coordination polymers. $Na^+$ was the most abundant metal ion in extracellular fluids. The advantages of Na lay in its cheap availability and good biocompatibility. As early as 1974, chemists began to use the physical properties of metal ions such as the size and hydrophilicity of alkali metal ions such as $Li^+$, $Na^+$, $K^+$, etc. to synthesize crown ether 15C5, 18C6, hole ligands C211, C222, etc. [41,42]. The compounds containing sodium were well water-soluble. Therefore, the derivatives of benzoic acid were chosen as the ligand of organic carboxylic acids, and the sodium ion as metal centre ion to synthesize a material.

So, this work was aimed to design a new material with both crystalline and fluorescent properties which may have great photostability and photobleaching resistance. Herein, a novel sodium-fluorescent crystal (Na-FS) was synthesized by 4-dimethylaminobenzoic acid and sodium hydroxide using one-pot hydrothermal method. Next, the structure of Na-FS was confirmed by single-crystal X-ray diffraction (XRD) and scanning electron microscope (SEM), the optical properties were studied by fluorescence spectrometer, and the thermal stability was analysed by thermogravimetric (TG) experiment.

# 2. Experimental section

## 2.1. Materials

4-dimethylaminobenzoic acid (purity: 97%) was purchased from Energy Chemical Co., Ltd (Shanghai, China), sodium hydroxide was purchased from Shanghai Wokai Biotechnology Co., Ltd. All chemicals

were used without further purification. Ultrapure water was obtained from a Millipore Milli-Q system (18.25 MΩ cm Thermo Scientific).

## 2.2. Apparatus and measurements

The single-crystal XRD data were on Bruker (APEX-II CCD, Germany) with Mo K$\alpha$ radiation ($\lambda$ = 0.71073 Å). SEM images of the synthesized crystal were obtained using SEM-Teneo VS (Teneo-VS, China). Fluorescence spectra were recorded on Cary Eclipse fluorescence spectrophotometer (Angilent Technologies, USA). The fluorescence lifetime of Na-FS was measured on fluorescence spectrofluorometers (FM-4P-TCSPC, USA). Fluorescence resistance to bleaching was obtained on Shimazu fluorescence spectrophotometer (RF-6000). The dissolved solution was ultrasonic by ultrasonic instrument (KQ-700E, China). The reaction liquid was concentrated with a rotary evaporator (R-100, Switzerland). TG of a powdered sample was performed in alumina crucible in the temperature range of 30–630°C, using a Mettler Toledo thermal analyser, at a heating rate of 10° min$^{-1}$.

## 2.3. Synthesis of Na-FS

Na-FS was synthesized by one-pot hydrothermal method. Firstly, 600 mg sodium hydroxide and 970 mg 4-dimethylaminobenzoic acid were dissolved in 15 ml ultrapure water in a beaker with stirring. After that, the mixed solution was treated with ultrasound for 20 min. Then, the obtained reaction solution was transferred into a 25 ml Teflon lined stainless autoclave and heated at 220°C for 24 h. After natural cooling to room temperature, the reaction solution was concentrated to about 4 ml by rotary evaporation, white crystals (Na-FS) quickly precipitated out, natural withering. Na-FS was stored in 5 ml EP tube at room temperature for further use.

## 2.4. Optical properties of Na-FS measurement

The optical properties of Na-FS were measured, such as fluorescence spectrum, fluorescence lifetime and fluorescence resistance to bleaching. Before testing, Na-FS was pulverized in a mortar until it was powdered and without sifted. The spectral test conditions were as follows: the voltage was 1000 V, the slit widths of the excitation and emission were both 10 nm. The fluorescence emission spectra were measured by fixed excitation wavelength on Cary Eclipse fluorescence spectrophotometer.

The fluorescence anti-bleaching test conditions were as described above. Briefly, the excitation wavelength was fixed at 290 nm and continuously excited Na-FS for 4200 s. Then, the fluorescence lifetime was acquired using a fluorescene spetrofluorometer (FM-4P-TCSPC, USA). Taking 290 nm excitation wavelength as an example, the test parameters were as follows: counts = 4000, 4095 channels, measurement range: 200 ns, which were used for analysis. Second-order fitting was used to process the data by origin; time was the product of time calibration and channel.

# 3. Results and discussion

## 3.1. The crystal structures

Details of the crystal parameters, data collection and refinements for Na-FS were summarized in table 1. Single-crystal X-ray data have been collected using Bruker APEX-II CCD diffractometer fitted with Mo K$\alpha$ radiation ($\lambda$ = 0.71073 Å). As the results depicted, the triclinic crystal system with space group of P-1 of the crystal and the estimated cell parameters are, $a$ = 10.5133(3) Å, $b$ = 15.9198(5) Å, $c$ = 15.9650(5) Å, $\alpha$ = 80.742(1)°, $\beta$ = 88.613(1)°, $\gamma$ = 83.507(1)° and $V$ = 2618.31(14) Å$^3$. Data collection was made at low temperature (170 (2) K) using the Crys Alis CCD software. Muti-scan for each image was used for data collection. The structures were solved by direct methods using SHELXT [43] and refined using SHEXL-2018 program [43]. The powder X-ray diffraction (PXRD) pattern of Na-FS is shown in figure 1; $2\theta$ of Na-FS are 12°, 20° and 26°, which had no standard XRD spectra in databases. That meant the developed crystal was a new type of crystal. And, the crystallographic data for the structures reported in this article have been deposited at the Cambridge Crystallographic Data Centre (CCDC) under the number of 2026042. This new crystal was named as 'Na-FS'.

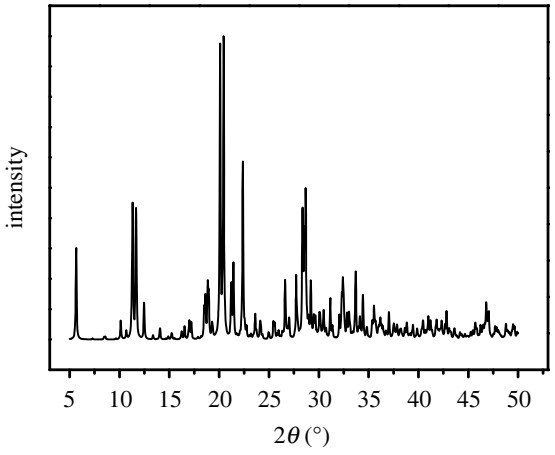

**Figure 1.** The powder X-ray diffraction (PXRD) pattern of Na-FS.

**Table 1.** Crystal data and structure refinement for Na-FS. (CCDC: 2026042, Na-FS).

| comp | parameter |
| --- | --- |
| chemical formula | $C_{36}H_{76}N_4Na_4O_{26}$ |
| MR (g mol$^{-1}$) | 1072.97 |
| temperature (K) | 170 |
| crystal system | triclinic |
| space group | P-1 |
| a, b, c (Å) | 10.5113(3), 15.9198(5), 15.9560(5) |
| cell ration | a/b = 0.6603, b/c = 0.977, c/a = 1.5180 |
| $\alpha$, $\beta$, $\gamma$ (deg) | 0.742(1), 88.613(1), 83.507(1) |
| V (Å$^3$) | 2618.31(14) |
| $\rho$ (g cm$^{-3}$) | 1.361 |
| Z | 2 |
| F(000) | 1144.00 |
| $\mu$ (mm$^{-1}$) | 0.141 |
| $T_{min}$, $T_{max}$ | 0.957, 0.972 |
| crystal size (mm) | $0.40 \times 0.26 \times 0.20$ |
| $\theta$ range for data collection (de) | 2.68–27.09 |
| index ranges | $-13 \leq h \leq 13$ |
| | $-20 \leq k \leq 20$ |
| | $-20 \leq l \leq 19$ |
| reflections collected | 32 050 |
| independent reflections ($R_{int}$) | 11 438 |
| reflections with $I > 2\sigma(I)$ | 9775 |
| data/restraints/parameter | 11 438/36/747 |
| goodness-of-fit on F2 | 1.017 |
| final R indexes ($I > 2\sigma(I)$) | R1 = 0.0409, wR2 = 0.0916 |
| final R indexes (all data) | R1 = 0.0330, wR2 = 0.0854 |
| largest difference in peak/hole (e Å$^3$) | 0.269/−0.291 |

The morphology of Na-FS was investigated in detail by SEM observation. The typical SEM image, as shown in figure 2*a*, gives was a general view of the morphology of the product over a large area in irregular laminar shape. Meanwhile, the size distribution of Na-FS is displayed in electronic

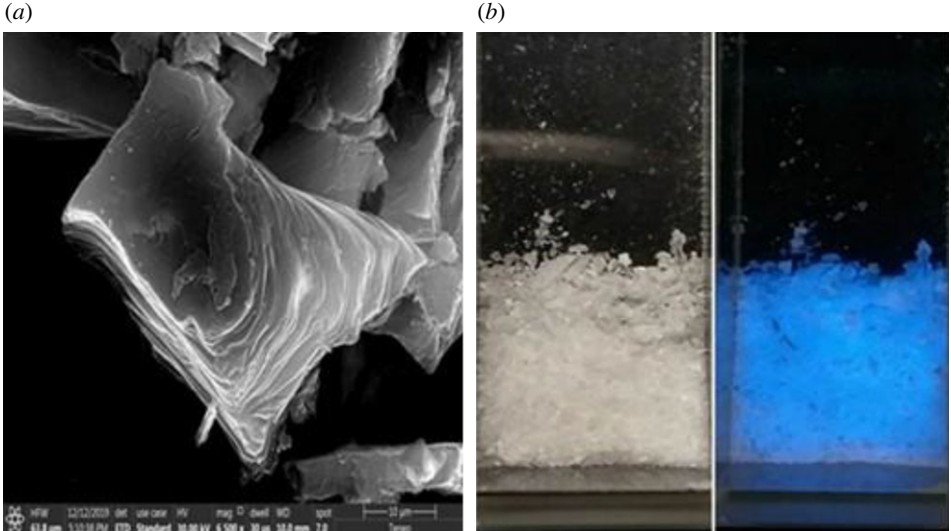

**Figure 2.** (*a*) SEM image of Na-FS; (*b*) the Na-FS the corresponding colours under 365 nm UV lamp (right) and daylight (left).

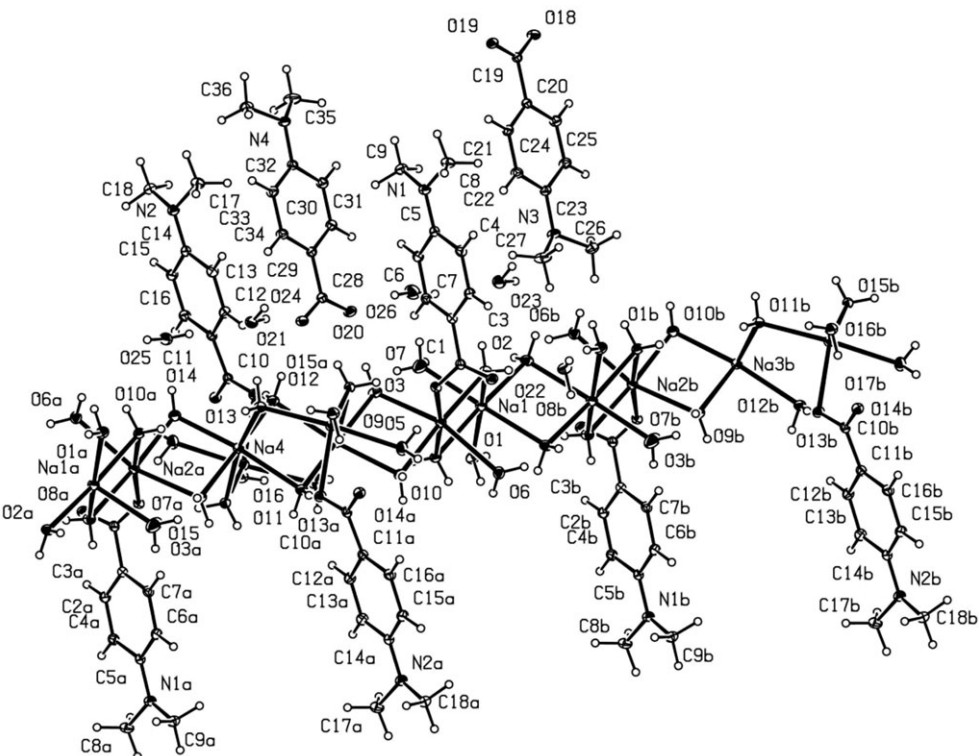

**Figure 3.** Crystal structure of Na-FS.

supplementary material, figure S1, which showed widely distribution with a diameter of 200−2000 nm. The morphology of Na-FS was also observed with the naked eye (figure 2*b*), it could be seen that the white crystal under the daylight and the white crystal had blue fluorescence emission under 360 nm UV lamp.

The molecular structure of the Na-FS is illustrated in figure 3. The central ion in the crystal structure was the sodium ion. The selected bond lengths and angles are given in table 2. The bond length of Na (1)–O (1) was 2.4661 (10) Å, and Na (1)–O (2) was 2.3941 (10) Å, the bond angle of O (11)–Na (3)–O (12i) was 94.14(3).

In a word, the results showed that a new crystal with blue fluorescence emission (Na-FS) was synthesized. The more details of this new material would discovered in further researches.

**Table 2.** Selected bond lengths (Å) and bond angle for Na-FS. (i) $1-x$, $1-y$, $1-z$; (ii) $2-x$, $1-y$, $2-z$.

| atoms 1,2 | bond lengths (Å) | atoms 1,2,3 | bond angle (°) |
|---|---|---|---|
| $Na_3Na_3^i$ | 3.4516(9) | $Na_3^iNa_3Na_4^i$ | 63.134(16) |
| $Na_3Na_2$ | 3.4253(7) | $Na_3^iNa_3Na_4$ | 62.413(15) |
| $Na_3Na_4^i$ | 3.7599(7) | $Na_2Na_3Na_3^i$ | 161.98(2) |
| $Na_3Na_4$ | 3.7843(7) | $Na_2Na_3Na_4^i$ | 99.487(16) |
| $Na_3O_{12}^i$ | 2.5014(10) | $Na_2Na_3Na_4$ | 134.716(17) |
| $Na_3O_{12}$ | 2.4180(9) | $Na_4iNa_3Na_4$ | 125.547(14) |
| $Na_3O_{15}^i$ | 2.5305(10) | $O_{12}Na_3Na_3^i$ | 46.44(2) |
| $Na_3O_{10}$ | 2.347(1) | $O_{12}iNa_3Na_{3i}$ | 44.46(2) |
| $Na_3O_{11}$ | 2.4072(10) | $O_{12}Na_3Na_2$ | 133.65(3) |
| $Na_3O_9$ | 2.3971(10) | $Na_4O_{11}Na_3$ | 104.60(4) |
| $Na_2Na_1$ | 3.5283(7) | $Na_4O_{11}H_{11A}$ | 111.1(13) |
| $Na_2O_7$ | 2.4201(9) | $Na_4O_{11}H_{11B}$ | 111.6(13) |
| $Na_2O_5$ | 2.4569(9) | $Na_3O_9Na_2$ | 90.71(3) |
| $Na_2O_6$ | 2.4692(10) | $Na_3O_9H_{9A}$ | 109.3(13) |
| $Na_2O_{10}$ | 2.3964(10) | $Na_3O_9H_{9B}$ | 126.3(14) |
| $Na_2O_9$ | 2.4172(10) | $Na_2O_9H_{9A}$ | 125.7(12) |
| $Na_2O_4$ | 2.3891(10) | $Na_2O_9H_{9B}$ | 103.9(13) |
| $Na_1Na_1^{ii}$ | 3.6350(9) | $Na_3O_{11}H_{11A}$ | 114.9(13) |
| $Na_1O_5$ | 2.4373(10) | $Na_3O_{11}H_{11B}$ | 113.4(13) |
| $Na_1O_2^{ii}$ | 2.4493(10) | $O_{11}Na_3O_{12}$ | 90.00(3) |
| $Na_1O_2$ | 2.3941(10) | $O_{11}Na_3O_{12}^i$ | 94.14(3) |
| $Na_1O_1$ | 2.4661(10) | $O_{11}Na_3O_{15}^i$ | 168.02(4) |
| $Na_1O_4$ | 2.4584(10) | $O_{11}Na_3O_{12}$ | 90.00(3) |
| $Na_1O_3$ | 2.3474(12) | $O_{11}Na_3O_{12}^i$ | 94.14(3) |
| $Na_4O_{15}$ | 2.3531(10) | $O_{11}Na_3O_{15i}$ | 168.02(4) |
| $Na_4O_{13}$ | 2.375(1) | $C_{14}Na_2C_{17}$ | 119.2(1) |
| $Na_4O_{11}$ | 2.3754(10) | $C_{18}Na_2C_{17}$ | 116.04(10) |
| $Na_4O_{17}$ | 2.3774(10) | $C_{32}Na_4C_{36}$ | 117.39(10) |
| $Na_4O_{16}$ | 2.3966(11) | $C_{32}Na_4C_{35}$ | 117.27(10) |
| $Na_4H_{16A}$ | 2.507(19) | $C_5Na_1C_8$ | 120.0(1) |
| $Na_4H_{12A}$ | 2.651(19) | $C_9Na_1C_8$ | 117.45(10) |

## 3.2. Thermogravimetric analysis

Thermal studies had been carried out using a Mettler Toledo thermal analyser. TG spectrum was recorded in the temperature range up to 630°C from 30°C, at a heating rate of $10°\ min^{-1}$. As shown in figure 4, a weight loss about 30% of Na-FS was observed in the first stage of TG curve (the temperature region from 100 to 140°C), which was due to the loss of 18 water molecules. In the second stage (140–400°C), the weight loss of about 20% was observed. It might be caused by the loss of incompact bonded components in Na-FS. At the temperature in range of 400–500°C, the weight of Na-FS was almost totally lost, which meant the overall disintegration and volatilization of the material. These results indicated that this material was unstable at high temperature, and it was suitable for room temperature to explore in later applications.

## 3.3. Fluorescence spectral characteristics of Na-FS

Fluorescence spectra of Na-FS are shown in figure 5. The fluorescence spectra of Na-FS exhibited an emission peak at 360 nm with an excitation range of 230–300 nm, and the quantum yield of the as-

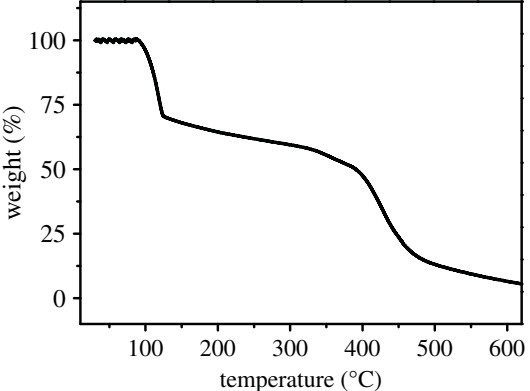

**Figure 4.** TG curve of the Na-FS.

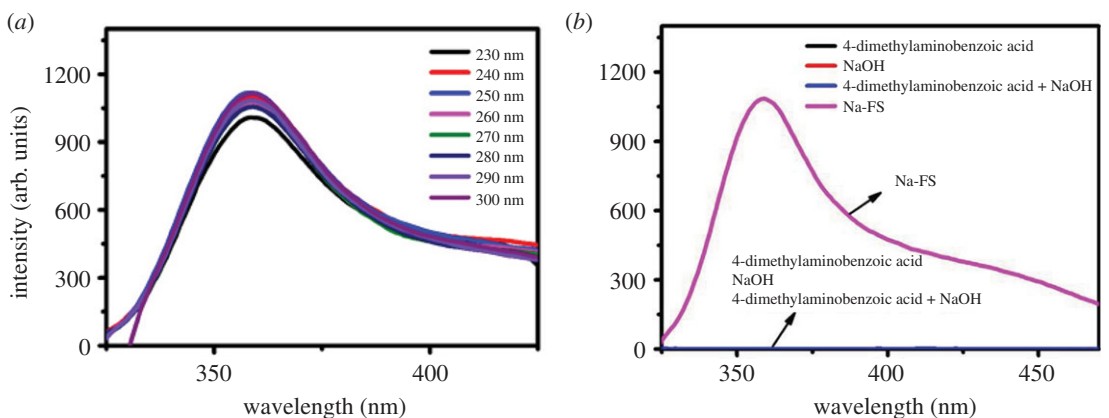

**Figure 5.** (*a*) The fluorescence spectra of Na-FS (excited at 230–300 nm). (*b*) The fluorescence spectra of Na-FS and raw materials.

prepared Na-FS is 30.58%. As shown in figure 5*a*, the Na-FS had a fluorescence emission peak at about 360 nm with a wide and continuous wavelength range, whose position remained unchanged along with the change of excitation wavelength. The results indicated that the Na-FS had an excitation-independent fluorescence behaviour, which might be attributed to their uniform particle size or differences in surface chemistry [44]. As shown in figure 5*b*, the reaction raw materials (4-dimethylaminobenzoic acid, sodium hydroxide and the mixture of them) had no fluorescence emission, only Na-FS had strong fluorescence emission. Fluorescence occurred for many reasons such as conjugated unsaturated bond enlargement, fluorescence resonance energy transfer (FRET), intramolecular charge transfer (ICT), photoinduced electron transfer (PET), ligand to ligand charge transfer (LLCT), ligand to metal charge transfer (LMCT) and so on. So our group hypothesized that the fluorescence of Na-FS might be due to the formation of a large conjugated structure, which caused the electrons to transfer to the empty d orbital of sodium ions [45], or in the absence of Na$^+$, the 4-dimethylaminobenzoic acid would work as an electron donor at the excited state, leading to low fluorescence quantum yield (PET) [46,47]. In the meanwhile, the PET process would be less effective upon coordination of Na$^+$, resulting in increased fluorescence.

## 3.4. Fluorescence lifetime characteristics of Na-FS

Based on the above fluorescence characteristics, we also studied the fluorescence lifetime of Na-FS. Taking the wavelength of 290 nm as an example, second-order fitting of the data was performed to obtain figure 6, and the results are shown in figure 5. The calculated mean life was 5.80 ns, $\chi^2$-value: 0.999.

## 3.5. Fluorescence stabilities of Na-FS

The fluorescence stabilities of Na-FS were studied in this part. For studying the effect of duration excitation time on fluorescence of Na-FS more deeply, the resistance to photobleaching of Na-FS was

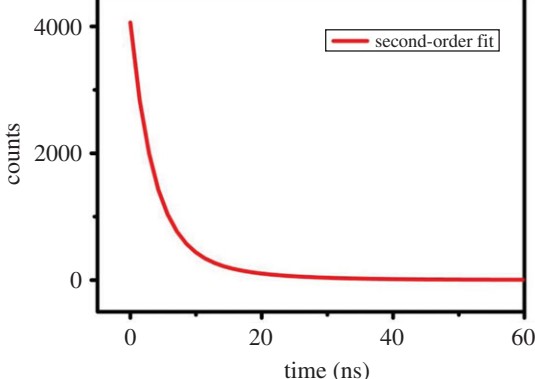

**Figure 6.** The fluorescence lifetime spectrum of the Na-FS.

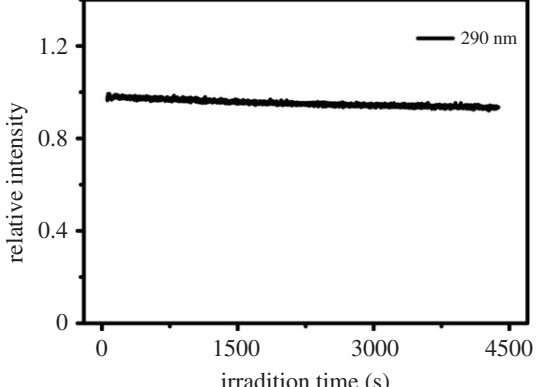

**Figure 7.** Fluorescence resistance to bleaching characteristics of Na-FS.

executed (figure 7). It was found that for the Na-FS under the excitation time of 4200 s at the wavelength of 290 nm, the fluorescence intensity remained unchanged, indicating that the Na-FS had good anti-bleaching property. It could be used to eliminate the bulk fluorescence of other substances in analytical biochemistry, which can eliminate the interference of other luminous materials. The stability of Na-FS fluorescence intensity in natural placement was also studied. As shown in electronic supplementary material, figure S2, the fluorescence intensity stayed at a steady value for 4 days. These results can proved that the developed crystal has excellent stability, which is beneficial for further application.

## 4. Conclusion

In conclusion, a novel Na-FS had been synthesized by one-pot hydrothermal method. The molecular structure of the Na-FS has been interpreted with single-crystal XRD study. Thermal studies were carried out to understand the thermal stability. The fluorescence properties elucidated the Na-FS emitted blue-green fluorescent, which can be excited in the range of 230–300 nm wavelength. The Na-FS had good photobleaching resistance and the average fluorescence life was 5.80 ns. As the excellent fluorescent properties of Na-FS, it was expected to have the more extensive applications, such as fluorescence detection of biomolecular ions and fluorescent falsification—proof.

Data accessibility. Crystallographic data for the structures reported in this article have been deposited at the Cambridge Crystallographic Data Centre under deposition number CCDC 2026042.

Authors' contributions. D.Y. designed the study and she is the corresponding author. Y.M. synthesized the Na-FS and helped draft the manuscript. Y.L. and Y.W. discovered its properties. F.Z. provided assistance for this work. All authors gave final approval for publication and agree to be held accountable for the work performed therein.

Competing interests. We declare we have no competing interests.

Funding. This work was financially supported by the National Science Foundation of China (grant no. 61901405), the Project of Science and Technology of Xuzhou Province (grant nos. KC19066 and KC18201) and Excellent Talents

Scientific Research Project (grant no. D2019026), Natural Science Foundation of Jiangsu Higher Education Institute of China (grant no. 19KJD350003) and China Postdoctoral Science Foundation (grant no. 2020M671608).

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
