## [Peer Review File · Royal Society Open Science]

Review History

RSOS-201987.R0 (Original submission)

Review form: Reviewer 1

Is the manuscript scientifically sound in its present form?

Yes

Are the interpretations and conclusions justified by the results?

Yes

Is the language acceptable?

Yes

Do you have any ethical concerns with this paper?

No

Have you any concerns about statistical analyses in this paper?

No

Recommendation?

Accept with minor revision (please list in comments)

Comments to the Author(s)

This is a quite straightforward study to synthesize a sodium-fluorescent crystal by a hydrothermal method. The crystal structure was measured by XRD, and the morphology was evaluated by SEM. The blue fluorescence was stable and resistant to photobleaching. I have a few suggestions for the authors below:

1. For the morphology evaluation, only SEM was used. More material characterization is suggested, for example, TEM, DLS etc.
2. How stable is Na-FS in the aqueous solution?
3. Why is Na-FS unique as a fluorescent material? Will it be more suitable for any particular applications? For example, cell labeling (imaging)?
4. What is the quantum yield for the fluorescence of Na-FS, e.g. excited by 290nm UV?

Review form: Reviewer 2

Is the manuscript scientifically sound in its present form?

Yes

Are the interpretations and conclusions justified by the results?

Yes

Is the language acceptable?

Yes

Do you have any ethical concerns with this paper?

No

Have you any concerns about statistical analyses in this paper?

No

Recommendation?

Major revision is needed (please make suggestions in comments)

Comments to the Author(s)

This paper reported a new sodium crystal that possesses unique fluorescent properties. The crystal was fully characterized, and its structure was interpreted. Overall, the paper is in decent shape. The experiments were well designed, and the results could support its hypothesis to some extent. I am very pleased to recommend its publication if the following questions could be properly addressed.

1. The authors claimed it is a "novel" sodium-fluorescent crystal. Did the authors do intensive enough research work to draw this conclusion? If it was the so-called "novel" one, the authors need to provide more details about its structure, say, the standard XRD peaks profile of this crystal. And it is also the obligation to name it, so future researchers can reproduce it if it is necessary to do so.
2. In figure 2, the authors used SRD to interpret the structure of the crystal. It is one of the most pivotal pieces of information to prove that the product is a real crystal. But the authors failed to reveal the information of one cell lattice. If the authors can provide that information in the manuscript, it will help the authors to understand the structure of the crystal better.
3. In figure 3, the authors employed TG to characterize the crystal, however, they did not fully interpret the data they got and failed to draw any conclusions based on the TG findings. Please revised that part in the manuscript accordingly.

4. In figure 4, please add the spectra of the ingredients of Na-FS to exclude the possibility that the fluorescence of Na-FS might come from one of its ingredients.
5. It will be better if the authors could find some practical application for these crystal materials.
6. Some typos need to be revised. I will only list some but not all. Page 4, line 15, "intramololular" should be "intramolecular"; Page 9, line 25, "rihgt" should be "right".

Decision letter (RSOS-201987.R0)

Dear Dr Ma:

Title: A Novel Sodium-Fluorescent Crystal
Manuscript ID: RSOS-201987

Thank you for submitting the above manuscript to Royal Society Open Science. On behalf of the Editors and the Royal Society of Chemistry, I am pleased to inform you that your manuscript will be accepted for publication in Royal Society Open Science subject to minor revision in accordance with the referee suggestions. Please find the reviewers' comments at the end of this email.

The reviewers and handling editors have recommended publication, but also suggest some minor revisions to your manuscript. Therefore, I invite you to respond to the comments and revise your manuscript.

Because the schedule for publication is very tight, it is a condition of publication that you submit the revised version of your manuscript before 22-Jan-2021. Please note that the revision deadline will expire at 00.00am on this date. If you do not think you will be able to meet this date please let me know immediately.

- 1) A text file of the manuscript (tex, txt, rtf, docx or doc), references, tables (including captions) and figure captions. Do not upload a PDF as your "Main Document".
- 2) A separate electronic file of each figure (EPS or print-quality PDF preferred (either format should be produced directly from original creation package), or original software format)

- 3) Included a 100 word media summary of your paper when requested at submission. Please ensure you have entered correct contact details (email, institution and telephone) in your user account
- 4) Included the raw data to support the claims made in your paper. You can either include your data as electronic supplementary material or upload to a repository and include the relevant doi within your manuscript
- 5) All supplementary materials accompanying an accepted article will be treated as in their final form. Note that the Royal Society will neither edit nor typeset supplementary material and it will be hosted as provided. Please ensure that the supplementary material includes the paper details where possible (authors, article title, journal name).

Kind regards,
Dr Laura Smith
Publishing Editor, Journals

On behalf of the Subject Editor Professor Anthony Stace and the Associate Editor Professor Tobias Hertel.

RSC Associate Editor:
Comments to the Author:
(There are no comments.)

RSC Subject Editor:
Comments to the Author:
(There are no comments.)

Reviewer comments to Author:
Reviewer: 1

Comments to the Author(s)
This is a quite straightforward study to synthesize a sodium-fluorescent crystal by a hydrothermal method. The crystal structure was measured by XRD, and the morphology was

evaluated by SEM. The blue fluorescence was stable and resistant to photobleaching. I have a few suggestions for the authors below:

1. For the morphology evaluation, only SEM was used. More material characterization is suggested, for example, TEM, DLS etc.
2. How stable is Na-FS in the aqueous solution?
3. Why is Na-FS unique as a fluorescent material? Will it be more suitable for any particular applications? For example, cell labeling (imaging)?
4. What is the quantum yield for the fluorescence of Na-FS, e.g. excited by 290nm UV?

Reviewer: 2

Comments to the Author(s)

This paper reported a new sodium crystal that possesses unique fluorescent properties. The crystal was fully characterized, and its structure was interpreted. Overall, the paper is in decent shape. The experiments were well designed, and the results could support its hypothesis to some extent. I am very pleased to recommend its publication if the following questions could be properly addressed.

1. The authors claimed it is a “novel” sodium-fluorescent crystal. Did the authors do intensive enough research work to draw this conclusion? If it was the so-called “novel” one, the authors need to provide more details about its structure, say, the standard XRD peaks profile of this crystal. And it is also the obligation to name it, so future researchers can reproduce it if it is necessary to do so.
2. In figure 2, the authors used SRD to interpret the structure of the crystal. It is one of the most pivotal pieces of information to prove that the product is a real crystal. But the authors failed to reveal the information of one cell lattice. If the authors can provide that information in the manuscript, it will help the authors to understand the structure of the crystal better.
3. In figure 3, the authors employed TG to characterize the crystal, however, they did not fully interpret the data they got and failed to draw any conclusions based on the TG findings. Please revised that part in the manuscript accordingly.
4. In figure 4, please add the spectra of the ingredients of Na-FS to exclude the possibility that the fluorescence of Na-FS might come from one of its ingredients.
5. It will be better if the authors could find some practical application for these crystal materials.
6. Some typos need to be revised. I will only list some but not all. Page 4, line 15, “intramololular” should be “intramolecular”; Page 9, line 25, “rihgt” should be “right”.

Author's Response to Decision Letter for (RSOS-201987.R0)

See Appendix A.

Decision letter (RSOS-201987.R1)

Dear Dr Ma:

Title: A Novel Sodium-Fluorescent Crystal

Manuscript ID: RSOS-201987.R1

It is a pleasure to accept your manuscript in its current form for publication in Royal Society Open Science. The chemistry content of Royal Society Open Science is published in collaboration with the Royal Society of Chemistry.

On behalf of the Subject Editor Professor Anthony Stace and the Associate Editor Professor Tobias Hertel.

RSC Associate Editor
Comments to the Author:
(There are no comments.)

Reviewer(s)' Comments to Author:

Appendix A

Response to Decision Letter

We highly appreciate the reviewers' kind consideration of the scientific content of our work. The comments and suggestions made by the reviewers are very helpful for us to revise the manuscript. Detailed reply to the comments and suggestions has been made as follows. (Note: The responses are highlighted in blue.)

Reviewer comments to Author:

Reviewer: 1

Comments to the Author(s)

This is a quite straightforward study to synthesize a sodium-fluorescent crystal by a hydrothermal method. The crystal structure was measured by XRD, and the morphology was evaluated by SEM. The blue fluorescence was stable and resistant to photobleaching. I have a few suggestions for the authors below:

1. For the morphology evaluation, only SEM was used. More material characterization is suggested, for example, TEM, DLS etc.

Answer: TEM characterization are unsuitable for this developed crystal, as the thicknesses of it (>200 nm) beyond the experimental range of TEM. So, the TEM image was not provided in this paper. The result of Na-FS size distribution was displayed in Fig.S1, which showed widely distribution with a diameter of 200–2000 nm. (Page 8)

2. How stable is Na-FS in the aqueous solution?

Answer: To answer this question, the stability of Na-FS fluorescence intensity in natural placement was studied. As shown in Fig.S2, the fluorescence intensity stay a steady value in 4 days. The fluorescence stability can prove Na-FS is a stable material in the aqueous solution. (Page S1 of supporting information).

3. Why is Na-FS unique as a fluorescent material? Will it be more suitable for any particular applications? For example, cell labeling (imaging)?

Answer: As the excellent fluorescent properties of Na-FS, it was expected to have the more extensive applications, such as fluorescence detection of biomolecular ions and fluorescent falsification – proof. (Page 14)

4. What is the quantum yield for the fluorescence of Na-FS, e.g. excited by 290nm UV?

Answer: The quantum yield of Na-FS is 30.58 % (when excited at 290nm). This part has been added in page 11 according to reviewer's kindly comments. The method detail

was provided in supporting information page S2.

Reviewer: 2

Comments to the Author(s)

This paper reported a new sodium crystal that possesses unique fluorescent properties. The crystal was fully characterized, and its structure was interpreted. Overall, the paper is in decent shape. The experiments were well designed, and the results could support its hypothesis to some extent. I am very pleased to recommend its publication if the following questions could be properly addressed.

1. The authors claimed it is a “novel” sodium-fluorescent crystal. Did the authors do intensive enough research work to draw this conclusion? If it was the so-called “novel” one, the authors need to provide more details about its structure, say, the standard XRD peaks profile of this crystal. And it is also the obligation to name it, so future researchers can reproduce it if it is necessary to do so.

Answer: The PXRD (powder X-ray diffraction) pattern of Na-FS was showed in Fig.1, 2θ of Na-FS are 12° , 20° and 26° , which had no standard XRD spectra in databases. That meant the developed crystal was a new type of crystal. And, the crystallographic data for the structures reported in this article have been deposited at the Cambridge Crystallographic Data Centre (CCDC) under the number of 2026042. This new crystal was named as “Na-FS”. (Page 6)

2. In figure 2, the authors used SRD to interpret the structure of the crystal. It is one of the most pivotal pieces of information to prove that the product is a real crystal. But the authors failed to reveal the information of one cell lattice. If the authors can provide that information in the manuscript, it will help the authors to understand the structure of the crystal better.

Answer: The cell parameters a, b and c of Na-FS was 10.5113(3), 15.9198(5), and 15.9560(5) Å, and the data has been displayed in Table 1 (Page 7). The Crystal Information (CIF) was uploaded to CCDC at the number of 2026042, which could be obtained free from CCDC if desired. (Page 14)

3. In figure 3, the authors employed TG to characterize the crystal, however, they did not fully interpret the data they got and failed to draw any conclusions based on the TG findings. Please revised that part in the manuscript accordingly.

Answer: TG characterization has been revised according to reviewer’s recommendation. As shown in Fig.4, a weight loss about 30% of Na-FS was observed in the first stage of TG curve (the temperature region from 100 °C to 140 °C), which was due to the loss of

18 water molecules. In the second stage (140 °C to 400 °C), the weight loss of about 20 % was observed. It might be caused by the loss of incompact bonded components in Na-FS. At the temperature in range of 400-500 °C, the weight of Na-FS was almost totally lost, which meant the overall disintegration and volatilization of the material. (Page 11)

4. In figure 4, please add the spectra of the ingredients of Na-FS to exclude the possibility that the fluorescence of Na-FS might come from one of its ingredients.

Answer: Yes, we agreed with reviewer's comments. The supplied data has been named as "Fig.5B". As shown, there is no fluorescence emission of 4-dimethylaminobenzoic acid, the mixture of 4-dimethylaminobenzoic acid and sodium hydroxide, and sodium hydroxide. (Page 11)

5. It will be better if the authors could find some practical application for these crystal materials.

Answer: As the excellent fluorescent properties of Na-FS, it was expected to have the more extensive applications, such as fluorescence detection of biomolecular ions and fluorescent falsification – proof. (Page 14)

6. Some typos need to be revised. I will only list some but not all. Page 4, line 15, "intramololular" should be "intramolecular"; Page 9, line 25, "rihgt" should be "right"

Answer: These problems in the manuscript had been corrected according to the review's recommendation.